# DAC: 2D-3D Retrieval with Noisy Labels via Divide-and-Conquer Alignment and Correction

## ABSTRACT

With the recent burst of 2D and 3D data, cross-modal retrieval has attracted increasing attention recently. However, manual labeling by non-experts will inevitably introduce corrupted annotations given ambiguous 2D/3D content, leading to performance degradation. Though previous works have addressed this issue by designing a naive division strategy with hand-crafted thresholds, their performance generally exhibits great sensitivity to the threshold value, implying their poor robustness in real-world scenarios. Besides, they fail to fully utilize the valuable supervisory signals within each divided subset. To tackle this problem, we propose a **D**ivide-and-conquer 2D-3D cross-modal **A**lignment and **C**orrection framework (DAC), which comprises **M**ultimodal **D**ynamic **D**ivision (MDD) and **A**daptive **A**lignment and **C**orrection (AAC). Specifically, the former performs accurate sample division by adaptive credibility modeling for each sample based on the compensation information within multimodal loss distribution. Then in AAC, samples in distinct subsets are exploited with different alignment strategies to fully enhance the semantic compactness and meanwhile alleviate overfitting to noisy labels, where a self-correction strategy is introduced to improve the quality of representation by mining the valuable supervisory signals from multimodal predictions as well. Moreover. To evaluate the effectiveness in real-world scenarios, we introduce a challenging noisy benchmark, namely Objaverse-N200, which comprises 200k-level samples annotated with 1156 realistic noisy labels. Extensive experiments on both traditional and the newly proposed benchmarks demonstrate the generality and superiority of our DAC, where DAC outperforms state-of-the-art models by a large margin (*i.e.*, with +5.9% gain on ModelNet40 and +5.8% on Objaverse-N200).

## CCS CONCEPTS

• **Computing methodologies → Computer vision representations**.

## KEYWORDS

2D-3D Retrieval, Divide-and-conquer, Label Correction

## 1 INTRODUCTION

With the rapid advancement of 3D acquisition technology, there has been a significant increase in the production and utilization of 3D

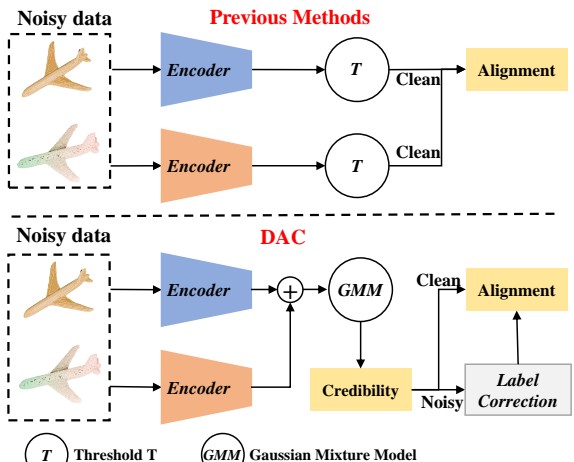

**Figure 1: Comparison between previous methods and our DAC. Our DAC employs a divide-and-conquer scheme to adaptively mine the discriminative semantics in distinct subsets, guided by the dynamically estimated credibility of each sample.**

data [8, 9]. An essential aspect of 3D data analysis and understanding is retrieving relevant 3D representations from given 2D/3D query input. This technique finds broad applications in virtual reality [13], autonomous driving [26, 32], and robotic manipulation [4, 34]. However, due to their complex geometrical structures, high dimensions, and irregular distribution, 3D data typically present more challenges compared to 2D data. Moreover, with the proliferation of large volumes of 3D data, data annotation is becoming increasingly expensive and time-consuming, leading to label noise issues in 2D-3D cross-modal retrieval. Consequently, learning with noisy labels has emerged as a crucial problem to address in 2D-3D cross-modal retrieval.

Previous methods for cross-modal retrieval could be categorized into two families: unsupervised-based and supervised-based. The former [12, 20, 21] focuses on aligning the cross-modal features from instance-based views, thus mitigating the inherent heterogeneity gap across different modalities. As for the latter ones[19, 22, 24, 43], they resort to a shared projection network for semantic gap reduction and adopt a traditional center loss to maximize the cross-modal correlation and minimize the intra-class variation. However, these methods generally suffer from significant performance degradation when applied in noisy scenarios. Thus, to tackle this issue, various methods are proposed to eliminate the negative impact of noise labels by designing Robust Clustering loss and employing sample division [12, 20].

However, the Robust Clustering loss [20] does not consider the sample-wise credibility and treats all the noise samples equally, which may fail in complex noisy scenarios. Inspired by the recent

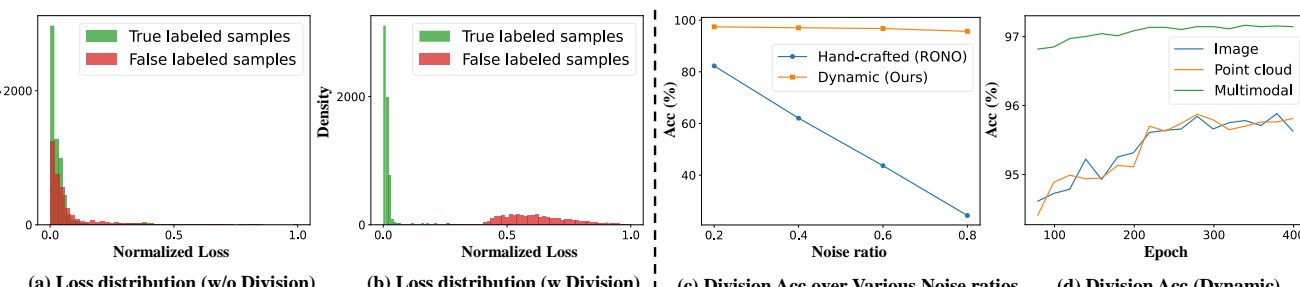

(a) Loss distribution (w/o Division)    (b) Loss distribution (w Division)    (c) Division Acc over Various Noise ratios    (d) Division Acc (Dynamic)

Figure 2: (a), (b) show the loss distribution of the noisy dataset after model convergence without/with sample division, respectively. Division Acc denotes the proportion of True and False labeled samples which is correctly identified by the sample division strategy. (c) show the Division Acc of Hand-crafted/Dynamic sample division strategies under different symmetric noise. (d) shows the division accuracy of Image-based/Point cloud-based/ Multimodal-based Dynamic sample division strategies in the training process. The experiments are conducted on the ModelNet40 under 40% symmetric noise.

progress of sample division based methods [6, 17, 27, 30] in Learning with Noisy Labels (LNL), we attempt to adopt a similar division scheme to adaptively utilize the inherent information within each sample. However, progress made by division strategies in 2D LNL may fail to be observed in cross-modal tasks due to the task-wise discrepancy. Thus we first investigate the impact of the division strategy on the cross-modal retrieval performance. Specifically, we visualize the loss distribution of training without/with sample division in Fig. 2 (a) and (b) respectively. It can be observed that models trained without sample division tend to over-fit on corrupted labels, exhibiting small losses for all false-labeled samples. Conversely, models trained with sample division showcase a distinct bimodal loss distribution. Such a phenomenon verifies the importance of sample division in noisy cross-modal retrieval.

Though the recent RONO [12] already integrates sample division to address the noisy issue, it adopts a simple hand-crafted threshold to distinguish true and false labeled samples as shown in Fig. 1. We empirically find that such a naive scheme showcases great sensitivity to parameter variations and lacks generalization in complex scenarios. Specifically, as shown in Fig. 2 (c), it can be observed that the division performance of RONO significantly deteriorates with the noise ratio increasing. Moreover, RONO performs sample division independently in each modality, and thus fails to utilize the potential complementary information of two modalities, leading to its inferior performance as shown in Fig. 2 (c). Additionally, to further verify the importance of the complementary effect, we conduct experiments by adopting the Gaussian Mixture Model (GMM) as the dynamic division strategy. As shown in Fig. 2 (d), unimodal-based strategies (i.e. Image or Point cloud), generally yield sub-optimal division accuracy due to the insufficient information within each modality.

To address the aforementioned problems, we propose a novel **D**ivide-and-conquer 2D-3D cross-modal **A**lignment and **C**orrection framework (DAC). Specifically, we introduce a **M**ultimodal **D**ynamic **D**ivision strategy(MDD), which adaptively captures the valuable noise-free semantic information in different modalities and constructs a multimodal loss distribution based on the cross-modal features. Subsequently, the credibility of each sample is adaptively modeled based on the multimodal loss distribution. With the credibility of each sample, we dynamically categorize the samples into clean and noisy sets. Then, we adopt a **A**daptive **A**lignment and

**C**orrection strategy(AAC) to conquer the samples in different subsets. Specifically, the samples in the clean set and noisy set are utilized for semantic and instance alignment respectively to eliminate the cross-modality gap while alleviating over-fitting to noisy labels. In addition, for the samples in the noisy set, a self-correction strategy is introduced to correct their corrupted labels with the model's multimodal predictions, which could further enhance the discrimination of representation. Moreover, we introduce a challenging realistic noisy benchmark: Objaverse-N200, which comprises 200k-level samples annotated with 1156 realistic noisy labels to evaluate the generalization ability of our model in real-world scenarios. In general, our contributions can be summarized as follows:

- We propose a novel and robust framework for 2D-3D cross-modal retrieval with noisy labels, namely DAC, which performs Divide-and-Conquer alignment for different noisy samples based on the dynamic-estimated credibility of each sample and adaptively utilize the supervisory information within different subsets.
- In DAC, a Multimodal Dynamic Division strategy (MDD) is proposed to dynamically model the credibility of each sample based on the multimodal loss distribution.
- In DAC, an Adaptive Alignment and Correction strategy (AAC) is designed for adaptive alignment of cross-modal features to fully harness the information and mitigate the negative impact of label noise. And an online correction scheme is designed to refurbish the corrupted label based on supervisory signals within the multimodal prediction.
- Our DAC demonstrates remarkable superiority over state-of-the-art methods on both traditional 3D object benchmarks with different scales of noisy labels and our newly constructed realistic benchmark: Objaverse-N200. In addition, it can be easily integrated with existing methods in a plug-and-play manner to boost their performance.

## 2 RELATED WORK

### 2.1 Cross-modal Retrieval.

Previous methods can be roughly categorized into two categories: unsupervised-based and supervised-based. Unsupervised-based approaches primarily focused on exploring the inherent structure

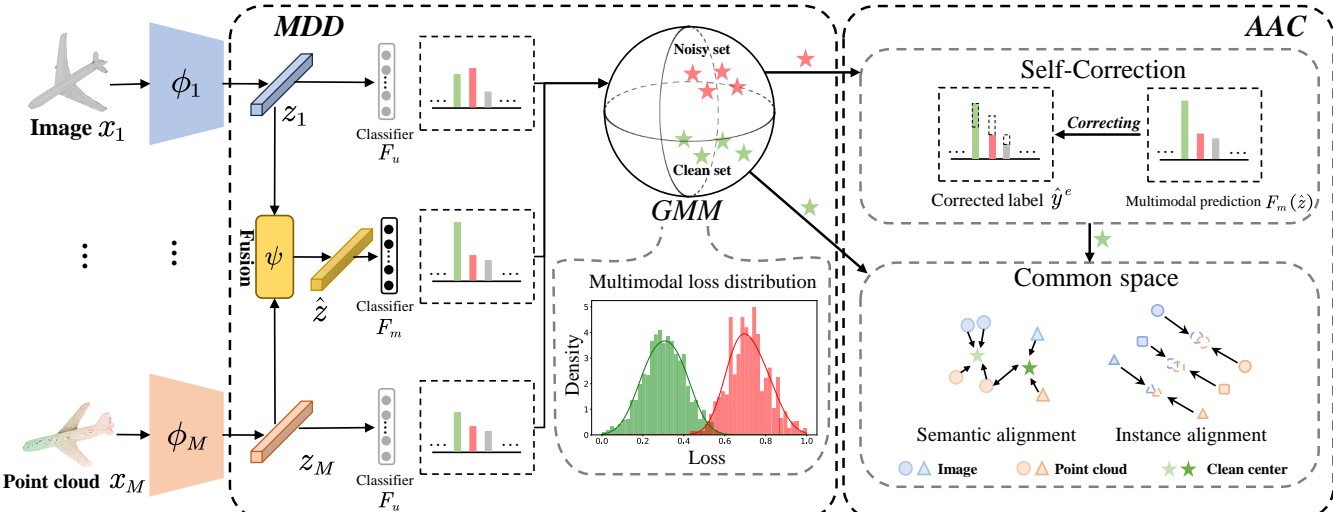

**Figure 3: An overview of our method DAC. (a) MDD: Multimodal Dynamic Division strategy (Sec. 3.2), (b) AAC: Adaptive Alignment and Correction strategy (Sec. 3.3). Our model performs Divide-and-Conquer alignment for different noisy samples based on the credibility of each sample. Specifically, MDD dynamically models the credibility of each sample based on the multimodal loss distribution of the dataset and divides the noisy samples into clean and noisy sets based on credibility. Then, AAC conquers different samples with adaptive alignment strategies and adopts a self-correction strategy to refurbish the corrupted label of samples.**

and relationships within multimodal data. Methods such as Canonical Correlation Analysis (CCA) [2, 21, 36, 42, 45] and Multi-View Learning [23, 29] were widely used to learn common representations across modalities. In addition, [20] adopts Multimodal Contrastive loss (MC) to maximize the mutual information between different modalities. On the other hand, supervised-based approaches leverage labeled data to explicitly learn the semantic correlations between modalities. [43] proposes to learn modality-invariant representations by both label and common representation supervision. Moreover, [22] utilize center loss to joint train the components of the cross-modal framework to find optimal features. However, most of the above works rely on well-labeled data, which cannot directly be transferred to noisy 2D-3D scenarios.

### 2.2 Learning with Noisy Labels.

Existing LNL methods could be roughly classified into robust architecture, robust loss, and sample selection approaches. The robust architecture-based methods focus on modeling the noise transition matrix with a noise adaptation layer [7, 15, 33]. However, these methods can not identify false-labeled examples, by treating all the samples equally. For robust loss-based methods, existing works [31, 37] leverage the self-prediction of the model to effectively refine the noisy labels to exploit the valuable semantics in the noise sample. Furthermore, various methods [39, 44] adopt an extra meta-corrector to obtain higher-quality corrected labels. However, they require a small clean dataset for evaluation, which is not realistic in real-world scenarios. Sample selection-based methods focus on selecting clean examples from a noisy dataset. Among them, the small loss trick is a popular selection criterion [6, 17, 30]. Based on the small-loss trick, DivideMix [27] adopts GMM to model the loss distribution, thus better distinguishing clean and noisy samples. However, these sample selection methods could not be directly

applied to the field of cross-modal retrieval due to the heterogeneity and geometric gap across 2D and 3D data. Additionally, when confronting complex and high-ratio noise scenarios, the GMM does not work well due to the largely overlapping of loss distribution and the confirmation bias issues [17].

## 3 METHODOLOGY

### 3.1 Overview

For convenience, we first clarify the notations within our DAC framework. Specifically, the noisy multimodal dataset is denoted as $\mathcal{D} = \{\mathcal{M}_j\}_{j=1}^M = \{\mathcal{X}_j, \mathcal{Y}_j\}_{j=1}^M$, where $M/K$ denote the number of modality and classes respectively. $\mathcal{M}_j$ can be represented as $\mathcal{M}_j = \{(x_i^j, y_i^j)\}_{i=1}^N$, where $N$ is the sample number, $x_i^j/y_i^j$ denote the $i$-th sample and noisy label in $\mathcal{M}_j$. The feature extractor for $\mathcal{M}_j$ is represented as $\phi_j$, and the feature $z_i^j = \phi(x_i^j)$. For simplicity, we denote the multimodal data pair for $i$-th sample as $(x_i, y_i) = (\{x_i^j\}_{j=1}^M, \{y_i^j\}_{j=1}^M)$.

Then we give out an overview of DAC framework. As shown in Fig. 3, DAC consists of two main components: Multimodal Dynamic Division (MDD) and Adaptive Alignment and Correction strategy (AAC). Specifically, the 2D and 3D samples in $\mathcal{D}$ are first fed into MDD to perform adaptive credibility modeling for each sample by considering multimodal loss distribution. Then these samples are divided into clean/noisy subsets based on the estimated credibility with adaptive thresholds. Afterward, in AAC, samples in the clean set are directly utilized for cross-modal alignment to fully harvest the discriminative information. For samples in the noisy set, we leverage the inherent supervisory signals in their multimodal predictions to perform label purification, which can mitigate the negative impact of noisy labels. Then the purified labels are used

for the same alignment strategy as the clean set to further enhance the representation compactness.

## 3.2 MDD: Multimodal Dynamic Division

To clarify our MDD more clearly, we first simply introduce the classification loss. For noisy sample $(x_i^j, y_i^j)$, to alleviate the semantic discrepancy across different modalities in label space, a classification loss is utilized to optimize the representation under label noise. Specifically, a shared classifier layer $F_u$ is proposed to obtain the classifier prediction $F_u(z_i^j; \theta_u)$ of each modality of the sample, and then a conventional cross-entropy (CE) loss is utilized to optimize the network, which is formulated as:

$$\mathcal{L}_{cls} = \frac{1}{MN} \sum_i^N \sum_j^M L(F_u(z_i^j; \theta_u), y_i^j) = -\frac{1}{MN} \sum_i^N \sum_j^M y_i^j \log([p_i^j]_{y_i^j}) \tag{1}$$

where, $p_i^j = \text{Softmax}(F_u(z_i^j; \theta_u))$ and $[p_i^j]_{y_i^j}$ is the $y_i^j$-th item of $p_i^j$. Then, we will explain how to construct the multimodal loss distribution based on the classification loss $L(F_u(z_i^j; \theta_u), y_i^j)$ in Eq. 1 and how to model the credibility of each sample based on the multimodal loss distribution in the remainder of this section.

**Multimodal loss distribution** Based on Eq. 1, we could model the unimodal loss distribution $\{L(F_u(z_i^j; \theta_u), y_i^j)\}_{(i=1, j=1)}^{(N,M)}$ of the noisy dataset. However, performing sample division based on the unimodal loss distribution overlooks the complementary information in the multimodal data, which is not sufficient to deal with complex noisy scenarios. In addition, unimodal loss distribution of true-labeled and false-labeled examples largely overlaps in complex noise scenarios. To tackle these issues, we propose a multimodal loss distribution, which could adaptively capture the valuable semantics in the multimodal data and generate a more discriminative distribution for sample division. Specifically, we first fuse the multimodal features $\{z_i^j\}_{j=1}^M$ for each sample $(x_i, y_i)$ by a fusion layer $\psi$, which is formulated as:

$$\hat{z}_i = \psi(z_i^1, ..., z_i^M) \tag{2}$$

where $\hat{z}_i$ is the fused feature. Then, a multimodal classifier $F_m$ is adopted to calculate the multimodal prediction $F_m(\hat{z}_i; \theta_m)$ for each noisy sample $(x_i, y_i)$ and the multimodal loss $l_i$ is calculated as:

$$l_i = L(F_m(\hat{z}_i; \theta_m), y_i) \tag{3}$$

To further avoid confirmation bias issues [17], we jointly utilize the shared classifier $F_u$ and the multimodal classifier $F_m$ to calculate the final multimodal loss distribution $l = \{l_i\}_{i=1}^N$, and the final multimodal loss $l_i$ is formulated as:

$$l_i = L(F_m(\hat{z}_i; \theta_m), y_i) + \frac{1}{M} \sum_j^M L\left(F_u\left(z_i^j; \theta_u\right), y_i^j\right) \tag{4}$$

Both $F_u$ and $F_m$ are optimized using Eq. 1.

**Sample credibility.** Due to the memorization effect of DNNs [3], clean samples usually converge at a faster pace than noisy ones, thus the multimodal loss distribution $l$ tends to be bimodal during the training stage. Hence, we adopt a two-component Gaussian Mixture Model (GMM) to fit the multimodal loss distribution $l$,

thereby estimating the credibility of each sample. The GMM is defined as:

$$p(l) = \sum_{t=1}^T \pi_t \mathcal{N}(l \mid \mu_t, \Sigma_t) \tag{5}$$

where $\mathcal{N}(l \mid \mu_t, \Sigma_t)$ is the Gaussian probability density function with mean $\mu_t$ and covariance $\Sigma_t$, and $\pi_t$ is the weight for $t$-th Gaussian component and we have $\sum_{t=1}^T \pi_t = 1$, and $T$ is the number of components of GMM. In our case, $T = 2$.

Then, we use the Expectation Maximization (EM) procedure [10] to fit the two components of GMM. Considering the efficiency, we iterate the EM procedure with 10 iterations to update the parameters(i.e. $\pi_t, \mu_t, \Sigma_t$) of GMM. Finally, we obtain the credibility $\gamma_i$ of each sample through the posterior probability:

$$\gamma_i = \frac{\pi_1 \mathcal{N}(l_i \mid \mu_1, \Sigma_1)}{\sum_{t=1}^2 \pi_t \mathcal{N}(l_i \mid \mu_t, \Sigma_t)} \tag{6}$$

where $\mathcal{N}(l_i \mid \mu_1, \Sigma_1)$ denotes the Gaussian component with a smaller mean (smaller loss).

With the adaptive credibility modeling, we could divide the noise data into clean set $S_c$ and noisy set $S_n$:

$$S_c = \{x_i | \gamma_i > \alpha\}, S_n = \{x_i | \gamma_i <= \alpha\} \tag{7}$$

where $\alpha$ is a threshold for clean-noise separation.

## 3.3 AAC: Adaptive Alignment and Correction.

Following the sample division, we employ adaptive alignment techniques to manage the distinct subsets, $S_c$ and $S_n$. To address the noisy samples in $S_n$, a self-correction mechanism is introduced, which leverages the model's multimodal predictions to refurbish the corrupted labels, thereby boosting the semantic compactness and discrimination of the learned representations.

**Adaptive Sample Alignment.** We employ different alignment strategies to deal with the samples in different subsets(i.e. $S_c$, $S_n$). For the samples in $S_c$, due to the high reliability of the labels, we directly use the labels for semantic alignment which could effectively mitigate the semantic gap across multimodal representations. Previous works adopt the center learning [12, 22] to achieve semantic alignment, which compacts the different modalities of samples to corresponding semantic centers in the common feature space. However, they optimize the center learning by a center loss in the form of absolute errors [12, 22], which is hard to optimize and limits the compactness of representation. To tackle these issues, we propose a contrastive center loss to optimize center learning, which is formulated as:

$$\mathcal{L}_{sem} = -\frac{1}{MN} \sum_{x_i \in S} \sum_j^M \log \left( \frac{e^{\frac{1}{\tau_c} \left(c_{(k=y_i^j)}\right)^T z_i^j}}{\sum_{n=1}^K e^{\frac{1}{\tau_c}(c_n)^T z_i^j}} \right), S = S_c / S_n \tag{8}$$

where $c_n$ is the learnable shared clustering center of $n$-th category in the common space, $\tau_c$ is a temperature parameter.

For the samples in $S_n$, due to the little discriminative information in their labels, we utilize them for training with instance alignment, which reduces the inherent gap between 2D and 3D data from the instance-based perspective. Specifically, we adopt a Multi-modal

**Table 1: Number of 3D models and categories in traditional datasets and Objaverse-N200.**

| Dataset | Objects | Classes |
|---|---|---|
| ModelNet10 [40] | 4899 | 10 |
| ModelNet40 [40] | 12311 | 40 |
| ShapeNet [5] | 51190 | 55 |
| ScanObjectNN [35] | 2902 | 15 |
| Objaverse-N200 | 194800 | 1156 |

Modal Gap loss(MG) [20] to optimize the cross-modal representation of samples, which is formulated as:

$$\mathcal{L}_{inst} = -\frac{1}{MN} \sum_{x_i \in S} \sum_{j}^{M} \log \left( \frac{\sum_{k}^{M} e^{\frac{1}{\tau_m} (z_i^k)^T z_i^j}}{\sum_{l}^{N} \sum_{m}^{M} e^{\frac{1}{\tau_m} (z_l^m)^T z_i^j}} \right), S = S_c/S_n \tag{9}$$

$\tau_m$ is a temperature parameter. Due to the randomness of the category centers $c_n$ caused by the random initialization, we also apply the instance alignment based on Eq. 9 for samples in $S_c$ to further alleviate the inherent gap across different modalities.

**Self-Correction(SC).** To exploit useful semantic information from the noisy set $S_n$, a self-correction strategy is proposed, which mines the valuable semantics from the classifier's self-prediction. Specifically, we utilize the classifier's prediction to correct the label of the sample in $S_n$. To obtain more reliable corrected labels, we correct the mislabeled samples based on the prediction of the multimodal classifier $F_m$. For each sample $(x_i, y_i)$ in $S_n$, the corrected label $\hat{y}_i$ is a soft label and is written as:

$$\hat{y}_i = \text{Softmax}(F_m(\hat{z}_i; \theta_m)) \tag{10}$$

However, due to the unstable training process, it is not reliable to refurbish the corrupted labels with the prediction of a single epoch. Therefore, we use the idea of Exponential Moving Average(EMA) to promote the reliability of our label correction. At epoch e, the moving-average corrected label over multiple training epochs is

$$\hat{y}_i^e = \mu \hat{y}_i^{(e-1)} + (1 - \mu) \hat{y}_i^e \tag{11}$$

where $\mu = 0.9$. In addition, for the corrected sample $(x_i, \arg\max(\hat{y}_i^e))$, we also employ semantic alignment based on Eq. 8 to further enhance the compactness of representation.

**Overall loss.** The complete loss function of our DAC can be formulated as:

$$\mathcal{L} = \mathcal{L}_{sem} + \lambda_1 \mathcal{L}_{inst} + \lambda_2 \mathcal{L}_{cls} \tag{12}$$

where $\lambda_1, \lambda_2$ are balance parameters for three losses.

## 4 OBJAVERSE-N200

To evaluate the effectiveness of our model in real-world scenarios, we developed a realistic and noisy benchmark, Objaverse-N200, built upon the recently proposed 3D dataset, Objaverse [9]. Due to the high expense and time-consuming nature of annotation, there is no category annotation for the object in Objaverse. To tackle this issue, we utilize a large pre-trained model Uni3D [46] to assign categories to these objects. Specifically, we first employ Uni3D for zero-shot classification, where unlabeled objects are assigned to their corresponding categories based on the category set derived from Objaverse-lvis. Then we collect the top 200 samples of each category to build the 3D dataset. Finally, we attain a real-world noisy

3D benchmark Objaverse-N200 containing about 200k 3D objects and annotated with 1156 realistic noisy labels, and the noise ratio of Objaverse-N200 is about 50%. The comparison between Objaverse-N200 and traditional 3D datasets is shown in table 1. Objaverse-N200 surpasses prior datasets by over an order of magnitude in size and number of categories. More information about Objaverse-N200 can be referred to our supplementary materials.

## 5 EXPERIMENTS

### 5.1 Experimental Setup

**Noisy test benchmarks.** We conduct extensive experiments on three traditional 3D datasets (3D MNIST [41], ModelNet10, ModelNet40 [40]) and our realistic noisy dataset Objaverse-N200. Following [12], we consider two noisy settings: Symmetric/Asymmetric noise, and use the same evaluation protocol. The noise rates for symmetric/asymmetric noise are [0.2, 0.4, 0.6, 0.8] and [0.1, 0.2, 0.4] respectively. It's important to note that Objaverse-N200 inherently constitutes a realistic noisy benchmark. Due to the space limitation, we only present the results of ModelNet10/40 and Objaverse-N200. More experimental results and details can be referred to our supplementary materials.

**Implementation details.** For the traditional datasets, we adopt the ResNet18 [18]/DGCNN [38]/MeshNet [11] as the backbone network for RGB/point cloud/mesh feature extraction. Then, all the features are projected to 256-D with two fully connected layers. The training epochs are 400 epochs with an initialized learning rate of 0.0001 and the learning rate dropped per 100 epochs with factor 10. For Objaverse-N200, we adopt the pre-trained image encoder and point cloud encoder in Uni3D [46] to extract the image and point cloud feature. Then, the features are projected to 1024-D with two fully connected layers. The training epochs are 40, while the initialized learning rate is 0.0001 and the learning rate is decayed with factor 10 at epochs 20 and 40. For all the datasets, the batch size is 128 and the optimizer is Adam [25] with a momentum of 0.9 for all the noisy benchmarks. $\tau_c$ and $\tau_m$ are set as 0.22 and 1.0 respectively. For the fusion layer $\psi$, we investigate various fusion structures (Add, Concat, Attention). Due to space limitation, the investigation of $\psi$ can be referred to our supplementary materials and we use the Concat (feature concatenation) as our final $\psi$.

### 5.2 Comparison with state-of-the-arts

**Results on traditional benchmarks.** To evaluate our method, we conduct detailed experiments on ModelNet10 and ModelNet40 [40] as shown in Tab. 2 and Tab. 3. As RONO employs a supervised pre-trained point cloud encoder [38] for feature extraction, which may bias the comparison, we also conduct experiments based on unsupervised pre-trained point cloud encoder [1], denoted by * in the results. And non-* denote using supervised pre-trained point cloud encoder, consistent with RONO's protocol [12]. From these results, we could obtain the following observations: 1) **Improvements on mAP.** Overall, our model(DAC/DAC*) achieved superior results compared to both unsupervised (DCCAE, UCCH, etc.) and supervised (MRL/MRL*, RONO/RONO*, etc.) methods across different noisy ratios. Specifically, our model achieves 5.8%↑ on ModelNet40 under 80% symmetric noise and 3%↑ on ModelNet40 under 20% asymmetric noise. 2) **Robustness.** The results reveal that previous

**Table 2: Performance comparison in terms of mAP from image to point cloud (Img → Pnt) and from point cloud to image (Pnt→Img) retrieval under the symmetric noise rates of 0.2, 0.4, 0.6, and 0.8 on the ModelNet10 and ModelNet40 datasets. ∗ denotes the use of an unsupervised pre-trained point cloud encoder [1]. The highest mAPs are shown in bold and the second highest mAPs are underlined.**

| Method | ModelNet10 [40] | | | | | | | | ModelNet40 [40] | | | | | | | |
|---|---|---|---|---|---|---|---|---|---|---|---|---|---|---|---|---|
| | Img → Pnt | | | | Pnt → Img | | | | Img → Pnt | | | | Pnt → Img | | | |
| | 0.2 | 0.4 | 0.6 | 0.8 | 0.2 | 0.4 | 0.6 | 0.8 | 0.2 | 0.4 | 0.6 | 0.8 | 0.2 | 0.4 | 0.6 | 0.8 |
| DCCAE [36] | 0.703 | 0.703 | 0.703 | 0.703 | 0.693 | 0.693 | 0.693 | 0.693 | 0.593 | 0.593 | 0.593 | 0.593 | 0.572 | 0.572 | 0.572 | 0.572 |
| DGCPN [42] | 0.765 | 0.765 | 0.765 | 0.765 | 0.759 | 0.759 | 0.759 | 0.759 | 0.705 | 0.705 | 0.705 | 0.705 | 0.697 | 0.697 | 0.697 | 0.697 |
| UCCH [21] | 0.771 | 0.771 | 0.771 | 0.771 | 0.770 | 0.770 | 0.770 | 0.770 | 0.755 | 0.755 | 0.755 | 0.755 | 0.739 | 0.739 | 0.739 | 0.739 |
| AGAH [16] | 0.853 | 0.736 | 0.583 | 0.425 | 0.837 | 0.699 | 0.549 | 0.408 | 0.809 | 0.732 | 0.687 | 0.568 | 0.783 | 0.736 | 0.664 | 0.554 |
| DSCMR [43] | 0.849 | 0.758 | 0.666 | 0.324 | 0.836 | 0.732 | 0.637 | 0.307 | 0.824 | 0.788 | 0.687 | 0.328 | 0.811 | 0.785 | 0.694 | 0.339 |
| MRL [20] | 0.876 | 0.870 | 0.863 | 0.832 | 0.861 | 0.857 | 0.848 | 0.823 | 0.833 | 0.829 | 0.828 | 0.818 | 0.824 | 0.826 | 0.820 | 0.817 |
| CLF [22] | 0.849 | 0.782 | 0.620 | 0.365 | 0.838 | 0.764 | 0.595 | 0.387 | 0.822 | 0.778 | 0.624 | 0.315 | 0.815 | 0.771 | 0.587 | 0.295 |
| CLF+MAE[14] | 0.853 | 0.752 | 0.679 | 0.343 | 0.838 | 0.716 | 0.659 | 0.373 | 0.827 | 0.758 | 0.651 | 0.384 | 0.816 | 0.749 | 0.640 | 0.372 |
| RONO [12] | 0.892 | 0.877 | 0.870 | 0.836 | 0.890 | 0.875 | 0.861 | 0.830 | 0.877 | 0.858 | 0.838 | 0.823 | 0.872 | 0.854 | 0.838 | 0.821 |
| DAC(Ours) | 0.898 | 0.897 | 0.890 | 0.879 | 0.901 | 0.895 | 0.888 | 0.881 | 0.894 | 0.893 | 0.893 | 0.879 | 0.886 | 0.885 | 0.884 | 0.871 |
| MRL* [20] | 0.880 | 0.859 | 0.786 | 0.687 | 0.975 | 0.849 | 0.764 | 0.684 | 0.808 | 0.784 | 0.746 | 0.573 | 0.802 | 0.773 | 0.731 | 0.535 |
| CLF* [22] | 0.861 | 0.761 | 0.510 | 0.247 | 0.870 | 0.768 | 0.512 | 0.267 | 0.811 | 0.710 | 0.467 | 0.143 | 0.822 | 0.758 | 0.520 | 0.229 |
| CLF*+MAE[14] | 0.868 | 0.763 | 0.542 | 0.259 | 0.867 | 0.792 | 0.520 | 0.268 | 0.803 | 0.680 | 0.682 | 0.151 | 0.803 | 0.721 | 0.555 | 0.129 |
| RONO* [12] | 0.915 | 0.905 | 0.890 | 0.859 | 0.911 | 0.892 | 0.871 | 0.835 | 0.876 | 0.861 | 0.844 | 0.791 | 0.872 | 0.860 | 0.841 | 0.788 |
| DAC*(Ours) | 0.920 | 0.920 | 0.899 | 0.886 | 0.920 | 0.915 | 0.899 | 0.883 | 0.887 | 0.885 | 0.872 | 0.849 | 0.884 | 0.882 | 0.867 | 0.847 |

**Table 3: Performance comparison under the asymmetric noise rates of 0.1, 0.2, and 0.4 on the ModelNet10 and ModelNet40 datasets. ∗ denotes the use of unsupervised pre-trained point cloud encoder [1].**

| Method | ModelNet10 [40] | | | | | | | | ModelNet40 [40] | | | | | | | |
|---|---|---|---|---|---|---|---|---|---|---|---|---|---|---|---|---|
| | Img → Pnt | | | | Pnt → Img | | | | Img → Pnt | | | | Pnt → Img | | | |
| | 0 | 0.1 | 0.2 | 0.4 | 0 | 0.1 | 0.2 | 0.4 | 0 | 0.1 | 0.2 | 0.4 | 0 | 0.1 | 0.2 | 0.4 |
| DCCAE [36] | 0.703 | 0.703 | 0.703 | 0.703 | 0.693 | 0.693 | 0.693 | 0.693 | 0.593 | 0.593 | 0.593 | 0.593 | 0.572 | 0.572 | 0.572 | 0.572 |
| DGCPN [42] | 0.765 | 0.765 | 0.765 | 0.765 | 0.759 | 0.759 | 0.759 | 0.759 | 0.705 | 0.705 | 0.705 | 0.705 | 0.697 | 0.697 | 0.697 | 0.697 |
| UCCH [21] | 0.771 | 0.771 | 0.771 | 0.771 | 0.770 | 0.770 | 0.770 | 0.770 | 0.755 | 0.755 | 0.755 | 0.755 | 0.739 | 0.739 | 0.739 | 0.739 |
| AGAH [16] | 0.862 | 0.821 | 0.805 | 0.756 | 0.867 | 0.827 | 0.801 | 0.743 | 0.807 | 0.817 | 0.778 | 0.778 | 0.799 | 0.800 | 0.779 | 0.761 |
| DSCMR [43] | 0.849 | 0.851 | 0.838 | 0.675 | 0.842 | 0.825 | 0.810 | 0.661 | 0.867 | 0.831 | 0.811 | 0.656 | 0.866 | 0.819 | 0.804 | 0.651 |
| MRL [20] | 0.887 | 0.869 | 0.867 | 0.859 | 0.871 | 0.865 | 0.861 | 0.854 | 0.848 | 0.846 | 0.838 | 0.811 | 0.843 | 0.844 | 0.838 | 0.799 |
| CLF [22] | 0.884 | 0.856 | 0.803 | 0.741 | 0.867 | 0.840 | 0.798 | 0.743 | 0.871 | 0.855 | 0.820 | 0.757 | 0.878 | 0.852 | 0.813 | 0.758 |
| CLF+MAE[14] | 0.877 | 0.848 | 0.794 | 0.771 | 0.853 | 0.841 | 0.791 | 0.754 | 0.864 | 0.837 | 0.811 | 0.761 | 0.853 | 0.832 | 0.798 | 0.763 |
| RONO[12] | 0.892 | 0.885 | 0.875 | 0.863 | 0.892 | 0.875 | 0.860 | 0.857 | 0.883 | 0.861 | 0.852 | 0.827 | 0.881 | 0.854 | 0.845 | 0.822 |
| DAC(Ours) | 0.906 | 0.904 | 0.904 | 0.883 | 0.904 | 0.900 | 0.902 | 0.886 | 0.895 | 0.893 | 0.893 | 0.871 | 0.893 | 0.888 | 0.888 | 0.871 |
| MRL* [20] | 0.890 | 0.889 | 0.880 | 0.740 | 0.888 | 0.885 | 0.868 | 0.734 | 0.848 | 0.814 | 0.793 | 0.710 | 0.843 | 0.800 | 0.787 | 0.717 |
| CLF* [22] | 0.884 | 0.880 | 0.838 | 0.675 | 0.867 | 0.875 | 0.829 | 0.638 | 0.871 | 0.832 | 0.799 | 0.745 | 0.878 | 0.834 | 0.799 | 0.751 |
| CLF*+MAE[14] | 0.883 | 0.878 | 0.844 | 0.679 | 0.880 | 0.876 | 0.829 | 0.644 | 0.864 | 0.830 | 0.818 | 750 | 0.853 | 0.829 | 0.817 | 0.749 |
| RONO*[12] | 0.924 | 0.917 | 0.915 | 0.891 | 0.924 | 0.913 | 0.909 | 0.891 | 0.886 | 0.861 | 0.859 | 0.824 | 0.885 | 0.856 | 0.850 | 0.830 |
| DAC*(Ours) | 0.925 | 0.925 | 0.924 | 0.897 | 0.924 | 0.924 | 0.924 | 0.899 | 0.889 | 0.887 | 0.887 | 0.847 | 0.887 | 0.882 | 0.880 | 0.848 |

methods, including RONO*, are particularly vulnerable to noise, especially at high noise levels, as evidenced by a 9.7% ↓ (88.5% to 78.8%) in performance on ModelNet40 with 80% symmetric noise. In contrast, our model exhibits exceptional robustness. Notably, DAC* maintains almost imperceptible performance degradation within 20%, 40%, and 60% symmetric noise. Remarkably, under 80% symmetric noise, our model only suffers 4% ↓ (88.9% to 84.9%). This highlights the stronger robustness of our model. 3) **Superiority even without label noise** Even without additional synthetic label noise, our model exhibits superior performance, demonstrating the effectiveness of our contrastive center loss. For clarity, the experiments on ModelNet10/ModelNet40 in the remainder of the paper employ unsupervised pre-trained point cloud encoder [1], unless otherwise explicitly stated. And we use DAC as our model, without distinguishing between DAC and DAC*.

**Results on Objaverse-N200.** To evaluate the robustness and effectiveness of our model in realistic scenarios, we conduct experiments

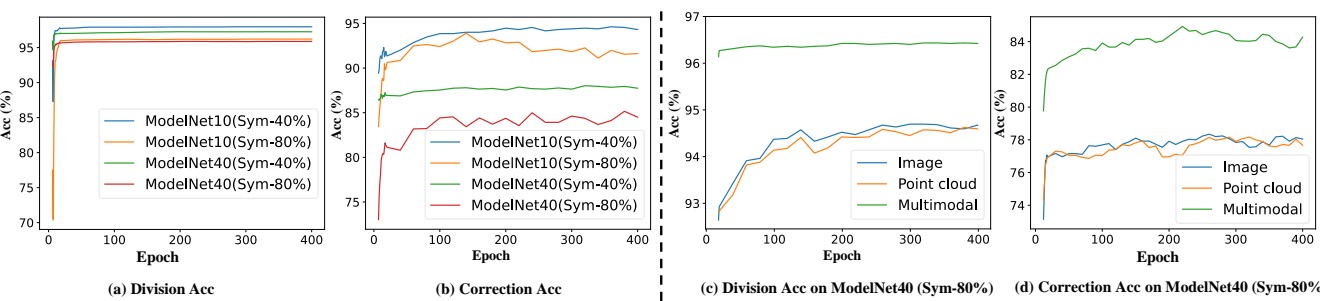

(a) Division Acc      (b) Correction Acc      (c) Division Acc on ModelNet40 (Sym-80%)      (d) Correction Acc on ModelNet40 (Sym-80%)

**Figure 4: Investigation of the division accuracy of MDD and the correction accuracy of the corrected labels generated by self-correction strategy on ModelNet10 and ModelNet40 under 40% symmetric noise (Sym-40%) and 80% symmetric noise (Sym-80%).**

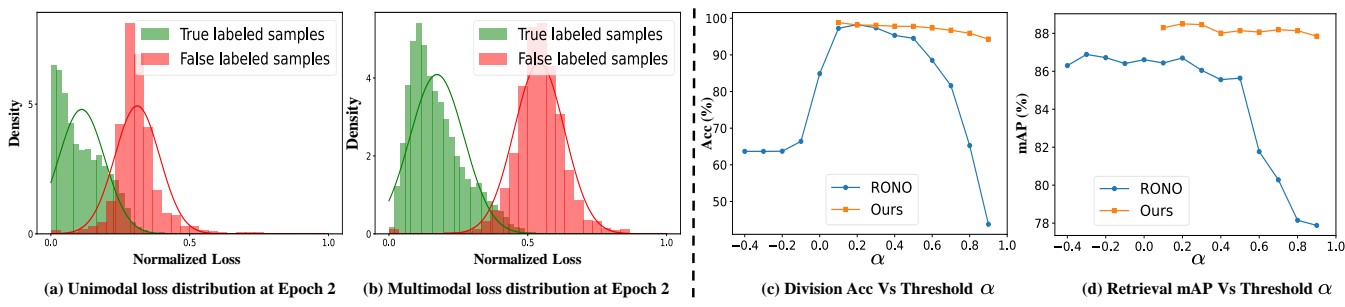

(a) Unimodal loss distribution at Epoch 2      (b) Multimodal loss distribution at Epoch 2      (c) Division Acc Vs Threshold $\alpha$      (d) Retrieval mAP Vs Threshold $\alpha$

**Figure 5: Investigation of the Multimodal loss distribution and division threshold $\alpha$ on ModelNet40 under 0.4 symmetric noise.**

**Table 4: Performance comparison between several popular cross-modal retrieval methods and their MDD application**

| Method | ModelNet10 | | | | ModelNet40 | | | |
| --- | --- | --- | --- | --- | --- | --- | --- | --- |
| | Img $\rightarrow$ Pnt | | Pnt $\rightarrow$ Img | | Img $\rightarrow$ Pnt | | Pnt $\rightarrow$ Img | |
| | 0.4 | 0.8 | 0.4 | 0.8 | 0.4 | 0.8 | 0.4 | 0.8 |
| MRL [20] | 0.859 | 0.687 | 0.849 | 0.684 | 0.784 | 0.573 | 0.773 | 0.535 |
| MRL + MDD | 0.892 | 0.815 | 0.887 | 0.862 | 0.842 | 0.759 | 0.857 | 0.743 |
| $\Delta(\%)$ | **+3.3** | **+12.8** | **+3.8** | **+17.8** | **+5.8** | **+18.6** | **+8.4** | **+20.8** |
| CLF [22] | 0.761 | 0.247 | 0.768 | 0.267 | 0.710 | 0.143 | 0.758 | 0.229 |
| CLF + MDD | 0.886 | 0.782 | 0.882 | 0.763 | 0.833 | 0.520 | 0.829 | 0.527 |
| $\Delta(\%)$ | **+12.5** | **+53.5** | **+11.4** | **+49.6** | **+12.3** | **+37.7** | **+7.1** | **+29.8** |
| RONO [12] | 0.905 | 0.859 | 0.892 | 0.835 | 0.861 | 0.791 | 0.860 | 0.788 |
| RONO + MDD | 0.910 | 0.871 | 0.901 | 0.852 | 0.874 | 0.815 | 0.870 | 0.812 |
| $\Delta(\%)$ | **+0.5** | **+1.2** | **+0.9** | **+1.7** | **+1.3** | **+2.4** | **+1.0** | **+2.4** |

**Table 5: Performance on the Objacerse-N200 datasets.**

| Method | Objaverse-N200 | |
| --- | --- | --- |
| | Img $\rightarrow$ Pnt | Pnt $\rightarrow$ Img |
| Openshape[28] | 0.274 | 0.257 |
| MRL[20] | 0.260 | 0.252 |
| CLF[22] | 0.238 | 0.192 |
| CLF[22]+MAE[14] | 0.237 | 0.192 |
| RONO[12] | 0.276 | 0.279 |
| Ours | **0.334** | **0.338** |

on the realistic noisy benchmark: Objaverse-N200. The comparison results are shown in Table 5. The results reveal that most prior methods yield lower performance compared to unsupervised approaches, indicating the complexity and challenge posed by realistic noise and these methods struggle to deal with the real-world noise. In contrast, our DAC model outperforms the unsupervised

**Table 6: Ablation studies for DAC on the on the ModelNet10 and ModelNet40 datasets with 0.4 and 0.8 symmetric noise.**

| MDD | | AAC | | | ModelNet10 [40] | | | | ModelNet40 [40] | | | |
| --- | --- | --- | --- | --- | --- | --- | --- | --- | --- | --- | --- | --- |
| $F_u$ | $F_m$ | $\mathcal{L}_{sem}$ | $\mathcal{L}_{inst}$ | SC | Img $\rightarrow$ Pnt | | Pnt $\rightarrow$ Img | | Img $\rightarrow$ Pnt | | Pnt $\rightarrow$ Img | |
| | | | | | 0.4 | 0.8 | 0.4 | 0.8 | 0.4 | 0.8 | 0.4 | 0.8 |
| $\checkmark$ | $\checkmark$ | $\checkmark$ | $\checkmark$ | $\checkmark$ | **0.920** | **0.886** | **0.915** | **0.883** | **0.882** | **0.849** | **0.878** | **0.847** |
| $\checkmark$ | | $\checkmark$ | $\checkmark$ | $\checkmark$ | 0.911 | 0.853 | 0.902 | 0.851 | 0.858 | 0.796 | 0.850 | 0.789 |
| | $\checkmark$ | $\checkmark$ | $\checkmark$ | $\checkmark$ | 0.915 | 0.881 | 0.910 | 0.880 | 0.880 | 0.843 | 0.877 | 0.837 |
| $\checkmark$ | $\checkmark$ | $\checkmark$ | $\checkmark$ | | 0.915 | 0.880 | 0.912 | 0.879 | 0.871 | 0.822 | 0.864 | 0.811 |
| $\checkmark$ | $\checkmark$ | $\checkmark$ | | | 0.913 | 0.879 | 0.910 | 0.875 | 0.867 | 0.812 | 0.859 | 0.803 |
| | | $\checkmark$ | $\checkmark$ | | 0.883 | 0.823 | 0.881 | 0.830 | 0.844 | 0.750 | 0.848 | 0.746 |
| | | | $\checkmark$ | | 0.869 | 0.746 | 0.867 | 0.741 | 0.820 | 0.657 | 0.822 | 0.640 |
| | | $\checkmark$ | | | 0.775 | 0.395 | 0.770 | 0.265 | 0.752 | 0.424 | 0.754 | 0.331 |

method openshape [28] significantly by 6.0%, which demonstrates the effectiveness of our model against real-world noise.

**Generality of MDD.** To validate the generality of our MDD, we integrate it with various popular traditional methods, including MRL [20], CLF [22], and RONO [12]. As shown in Table 4, MDD consistently brings substantial improvements across various noise ratios for these methods. Notably, MDD boosts CLF's performance by an impressive 49.6% on ModelNet10 under 80% symmetric noise, which demonstrates its superior capacity to handle severe noise. Consequently, these results confirm the generality of our MDD in enhancing existing methodologies.

### 5.3 More analysis

**Investigation on Division and Self-Correction.** We evaluate the effectiveness of our proposed division strategy, MDD, and self-correction strategy in AAC in Fig. 4. As depicted in 4(a) and (b), our methods consistently exhibit high accuracy in both dividing and correcting noisy samples across distinct datasets with varying noise

**Table 7: Performance comparison of RONO [12] and our DAC on trimodal ModelNet40 dataset [40]. For a convenience presentation, we abbreviate Image, Mesh, Point cloud, Query, and Retrieval to Img, Msh, Pnt, Qry, and Retrv, respectively.**

| $\eta$ | Qry | Img | | | Msh | | | Pnt | | |
|---|---|---|---|---|---|---|---|---|---|---|
| | Retrv | Img | Msh | Pnt | Img | Msh | Pnt | Img | Msh | Pnt |
| 0 | RONO | 0.911 | 0.901 | 0.891 | 0.899 | 0.901 | 0.883 | 0.891 | 0.894 | 0.891 |
| | Ours | **0.915** | **0.905** | **0.893** | **0.903** | **0.902** | **0.893** | **0.895** | **0.895** | **0.896** |
| 0.2 | RONO | 0.874 | 0.872 | 0.881 | 0.883 | 0.891 | 0.889 | 0.875 | 0.884 | 0.890 |
| | Ours | **0.899** | **0.893** | **0.890** | **0.884** | **0.892** | **0.891** | **0.884** | **0.887** | **0.892** |
| 0.4 | RONO | 0.858 | 0.876 | 0.863 | 0.862 | 0.881 | 0.875 | 0.859 | 0.875 | 0.875 |
| | Ours | **0.898** | **0.900** | **0.892** | **0.893** | **0.901** | **0.890** | **0.883** | **0.890** | **0.886** |
| 0.6 | RONO | 0.842 | 0.853 | 0.851 | 0.857 | 0.857 | 0.862 | 0.843 | 0.868 | 0.872 |
| | Ours | **0.890** | **0.891** | **0.886** | **0.882** | **0.889** | **0.883** | **0.880** | **0.884** | **0.884** |
| 0.8 | RONO | 0.828 | 0.842 | 0.842 | 0.842 | 0.868 | 0.866 | 0.841 | 0.864 | 0.868 |
| | Ours | **0.889** | **0.883** | **0.885** | **0.878** | **0.877** | **0.876** | **0.880** | **0.876** | **0.882** |

ratios. Notably, MDD attains division accuracy above 95% across all noise levels, demonstrating its effectiveness and robustness. Additionally, we investigate the impact of multimodal information on division and correction, as depicted in 4(c) and (d). The integration of multimodal features brings performance improvement for sample division and label correction, which is attributed to the inherent richness of multimodal data, including depth insights from 3D data and multi-view information from 2D images. These findings substantiate the complementary nature of distinct modalities.

**Investigation on Multimodal loss distribution.** We present experimental results to investigate our proposed multimodal loss distribution in Fig. 5 (a) and (b). Our results revealed the following key insights: 1) Mitigating overfitting of unimodal data. By comparing the unimodal and multimodal loss distributions, we noticed that false-labeled samples typically exhibit lower losses in the unimodal case, which can lead to overfitting and confirmation bias. Conversely, in the multimodal loss distribution, the normalized loss of these false-labeled samples is generally above 0.5, indicating a higher level of robustness 2) Enhancing reliability of sample division. The multimodal loss distribution exhibits a smaller overlap between its components compared to the unimodal one, indicating a higher reliability in sample division. This is attributed to the effective fusion of complementary information from different modalities. More experiments of loss distribution can be referred to our supplementary materials.

**Ablation Study.** To assess the effectiveness of MDD and AAC, we conducted detailed ablation studies on ModelNet10 and ModelNet40, as presented in Table 6. Comparing row 1 with rows 6, 7, and 8, adopting MDD to divide the noisy dataset before alignment significantly boosts model performance (9.9% and 10.1% ↑ for Sym-4% and Sym-80%), highlighting the effectiveness and robustness of our MDD. By comparing row 1 with row 2 and row 3, we observe that it is insufficient to divide the noisy dataset based on unimodal data. When adopting the fused multimodal feature with $F_m$, both the performance of ModelNet10 and ModelNet40 [40] can be improved significantly, especially under serve noisy scenes, due to its adaptive capture of semantics. Moreover, combining both $F_u$ and $F_m$ leads to additional performance gains (6.8% ↑ on ModelNet40 (Sym-80%)) as it addresses the confirmation bias of the single classifier($F_u/F_m$). Additionally, our AAC could further

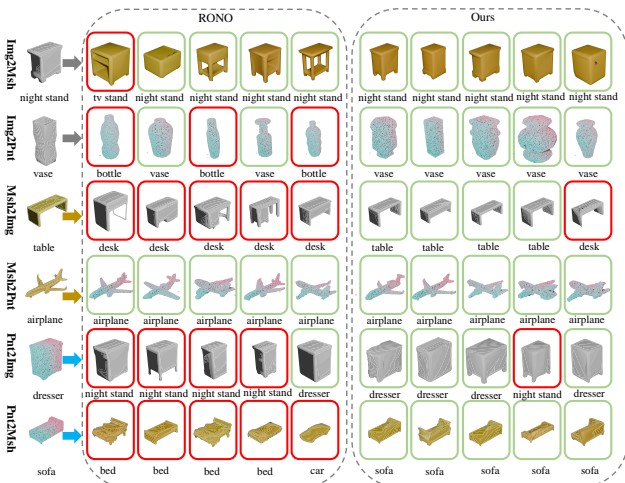

**Figure 6: Top-5 retrieved results of RONO and our DAC under 0.4 symmetric noise on the ModelNet40 dataset. Green/red boxes indicate correct/wrong retrieval respectively.**

improve model performance (3.6% on ModelNet40 with 80% symmetric noise), evidencing the effectiveness of adaptive alignment and the reliability of corrected labels. These results demonstrate the effectiveness of each component.

**Parameter sensitivity.** To evaluate the influence of threshold $\alpha$ on RONO [12] and our DAC, we conducted experiments on ModelNet40 [40] with 40% symmetric noise. As shown in Fig. 5 (c) and (d), RONO [12] demonstrates significant sensitivity to $\alpha$, resulting in poor robustness and sub-optimal performance. Conversely, our DAC exhibits superior and stable performance across various $\alpha$ values, indicating its robustness to the threshold $\alpha$. And, we set $\alpha$ as 0.5 in our experiments.

**Further comparison with RONO.** To thoroughly evaluate the performance of our proposed DAC, we conducted extensive retrieval and visualization experiments across three modalities (image, point cloud, and mesh) on ModelNet10 and ModelNet40 [40], comparing it with the state-of-the-art RONO [12]. The results, presented in Tab.7 and Fig. 6, show that our DAC consistently outperforms RONO across various noisy ratios, demonstrating the robustness and effectiveness of our DAC.

## 6 CONCLUSION

In this paper, we propose a novel robust Divide-and-conquer Alignment and Correction framework (DAC) for noisy 2D-3D cross-modal retrieval. Specifically, our DAC employs a novel Multimodal Dynamic Division strategy to dynamically divide the noisy dataset based on the multimodal loss distribution. With the help of division, we design an Adaptive Alignment and Correction strategy to enhance the semantic compactness of representation while mining the implicit semantics in noisy samples. In addition, we introduce a challenging realistic noisy benchmark: Objaverse-N200. We conduct extensive experiments on both traditional 3D model datasets and Objaverse-N200 to demonstrate the superior robustness and effectiveness of our DAC against synthetic and realistic noise. Our method also can be integrated with existing methods as a plug-and-play strategy to achieve further improvement.

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
