# OpenReview forum: "DAC: 2D-3D Retrieval with Noisy Labels via Divide-and-Conquer Alignment and Correction"
_acmmm.org/ACMMM/2024/Conference — MM2024 Poster_

### Official Review · Reviewer_A2Mt · 2024-05-02

**Rating:** 3
**Confidence:** 3

**Summary:**

The paper studies cross-modal retrieval with noisy label. Compared to previous approach, the main contribution is to adopt the posterior probability of GMM to judge the real or fake sample label, which is called as Divide-and-Conquer alignment for different noisy samples.

Extensive experiments on both traditional and the newly proposed benchmarks demonstrate the effectiveness.

**Strengths:**

1. The paper introduces the posterior probability of GMM to divide the samle into two sets, which is S_c and S_n.
2. Experiments is ok for me, especially on the introduced challenging realistic noisy benchmark: Objaverse-N200.

**Limitations:**

1. The upper and lower subscripts in the main text of the paper and the upper and lower subscripts in the figure are represented differently, which can easily cause confusion during reading.
2. The S in formula 8 is unclear.
3. The fusion layer is learnable or not? From experiment, it seems only adopting Concat, which is better than attention and so on?
4. The process of using GMM model to determine the authenticity of labels is something like hard assignment, why dont consider about soft assignment?

**Suitability:**

2

---

### Official Review · Reviewer_deLx · 2024-05-16

**Rating:** 5
**Confidence:** 3

**Summary:**

This paper investigates the problem of 2D-3D retrieval with noisy labels and provide a divide-and-conquer alignment and correction for this problem.

**Strengths:**

1. The motivation is clear.
2. The methodology is reasonable.
3. The experiments are thorough.

**Limitations:**

The divide-and-conquer strategy resembles some existing works. Please provide a discussion with them.

**Suitability:**

3

---

### Official Review · Reviewer_Rx5c · 2024-05-21

**Rating:** 5
**Confidence:** 4

**Summary:**

Past approaches with fixed thresholds for data division show limited robustness.
The Divide-and-Conquer 2D-3D Alignment and Correction (DAC) framework introduces Multimodal Dynamic Division (MDD) for adaptive sample division based on loss compensation, and Adaptive Alignment and Correction (AAC) to refine representations via subset-specific strategies and self-correction, combating noisy labels.
To validate real-world performance, the authors present Objaverse-N200, a benchmark with 200,000 samples and 1156 noisy labels, where DAC demonstrates remarkable superiority, outperforming current models significantly.

**Strengths:**

The author's ideas are simple and easy to understand
The writing is clear.
The experiments are relatively sufficient and prove the effectiveness of the method.

**Limitations:**

There are several issues that need to be addressed below.

1. In line 493, why can the multimodal classifier correct wrong labels? If the multimodal classifier does not classify well enough, does it correct the error? Is the multimodal classifier actually a teacher model?
2. Can the Gaussian component be set to T=3 or T=4?
3. This algorithm requires 10 iterations to update the GMM parameters. Will this cause the training and testing time of DAC to be higher than that of RONO?
4. For pseudo-label generation technology, more related work still needs to be discussed. For example [1][2].

[1] DistillHash: Unsupervised Deep Hashing by Distilling Data Pairs
[2] Dual-Stream Knowledge-Preserving Hashing for Unsupervised Video Retrieval

**Suitability:**

3

---

### Official Review · Reviewer_eHyR · 2024-05-24

**Rating:** 4
**Confidence:** 2

**Summary:**

This paper addresses the challenges in cross-modal retrieval of 2D and 3D data, specifically focusing on performance degradation due to corrupted annotations from non-expert labeling. Existing methods using naive division strategies and hand-crafted thresholds show poor robustness and fail to fully utilize supervisory signals. The authors propose the Divide-and-Conquer 2D-3D cross-modal Alignment and Correction framework (DAC), which includes Multimodal Dynamic Division (MDD) for accurate sample division and Adaptive Alignment and Correction (AAC) for enhancing semantic compactness and mitigating overfitting. DAC also introduces a self-correction strategy to improve representation quality. To test their approach, the authors introduce a new benchmark, Objaverse-N200. Extensive experiments show that DAC significantly outperforms state-of-the-art models.

**Strengths:**

1. The proposed MDD and AAC methods make sense and appear to be effective.

2. The experiments are comprehensive, effectively showing the superiority of the proposed method.

**Limitations:**

1 The discussion on methods related to 2D-3D retrieval is missing in the related work section, which is crucial for the completeness of the paper. If it is mentioned in the section, it should be explicitly highlighted.

2 There are several areas that can be improved. For instance, punctuation is needed after formulas, there is an extra punctuation mark in the title of Section 3.3, the title of Table 4 lacks punctuation, and the notation "𝑀/𝐾" on line 327 looks like division.

3 I am still unclear about the necessity of using GMM. It would be helpful to provide more explanation on this aspect.

**Suitability:**

3

---

### Meta-Review · Area_Chair_4Yp5 · 2024-06-28

**Recommendation:** Accept (Poster)
**Confidence:** 3

**Metareview:**

All four reviewers agree that this paper introduces a new benchmark, Objaverse-N200, and clearly articulates its merit and originality. The paper is well-written, and the experiments are relatively sufficient, effectively demonstrating the method's effectiveness. Based on these points, I recommend acceptance.